# Accuracy of Thoracic Ultrasonography for the Diagnosis of Pediatric Pneumonia: A Systematic Review and Meta-Analysis

**DOI:** 10.3390/diagnostics13223457

**Published:** 2023-11-16

**Authors:** Zhenghao Dong, Cheng Shen, Jinhai Tang, Beinuo Wang, Hu Liao

**Affiliations:** 1Department of Thoracic Surgery, West China Hospital, Sichuan University, Chengdu 610041, China; dzh990913@163.com (Z.D.); shencheng@scu.edu.cn (C.S.); wang_beinuo@163.com (B.W.); 2Department of Radiation Oncology, The First Affiliated Hospital of Dalian Medical University, Dalian 116011, China

**Keywords:** thoracic ultrasonography, chest radiography, pneumonia, diagnostic accuracy, ultrasonic diagnosis

## Abstract

As an emerging imaging technique, thoracic ultrasonography (TUS) is increasingly utilized in the diagnosis of lung diseases in children and newborns, especially in emergency and critical settings. This systematic review aimed to estimate the diagnostic accuracy of TUS in childhood pneumonia. We searched Embase, PubMed, and Web of Science for studies until July 2023 using both TUS and chest radiography (CR) for the diagnosis of pediatric pneumonia. Two researchers independently screened the literature based on the inclusion and exclusion criteria, collected the results, and assessed the risk of bias using the Diagnostic Accuracy Study Quality Assessment (QUADAS) tool. A total of 26 articles met our inclusion criteria and were included in the final analysis, including 22 prospective studies and four retrospective studies. The StataMP 14.0 software was used for the analysis of the study. The overall pooled sensitivity was 0.95 [95% confidence intervals (CI), 0.92–0.97] and the specificity was 0.94 [95% CI, 0.88–0.97], depicting a good diagnostic accuracy. Our results indicated that TUS was an effective imaging modality for detecting pediatric pneumonia. It is a potential alternative to CXR and a follow-up for pediatric pneumonia due to its simplicity, versatility, low cost, and lack of radiation hazards.

## 1. Introduction

As a common lower respiratory tract infection, pneumonia is one of the important causes of morbidity and mortality in children. According to statistics, around 150 million children under the age of five are afflicted with pneumonia annually, and pneumonia-related mortality accounts for approximately 15% of the overall mortality in this age cohort [1,2]. Accurate and sensitive diagnoses of pediatric pneumonia can help clinicians implement more effective therapies to prevent further progression of the condition. The symptoms and signs of pneumonia in children, however, lack specificity and often exhibit variations based on the patient’s age, pathogen type, and infection severity. Therefore, more rapid and efficient ways to diagnose pediatric pneumonia still need to be further explored [3].

Currently, CXR is considered the primary diagnostic tool for suspected pneumonia in children, while chest computed tomography (CT) is frequently employed as the definitive diagnostic method for patients with inconclusive CXR findings [4,5]. However, despite their high diagnostic accuracy for identifying the etiology of chest opacity, these aforementioned imaging modalities are not recommended as the first-line choice for pediatric pneumonia evaluation due to their high cost, demanding operator and equipment requirements, prolonged patient waiting times, and potential exposure to hazardous levels of radiation for children [6,7,8].

In recent years, there has been increasing interest in exploring thoracic ultrasonography (TUS) as a potential alternative option for the diagnosis of pediatric pneumonia, owing to the continuous advancements in ultrasound technology. Compared to CT and CXR, TUS offers no ionizing radiation exposure, low cost, and ease of operation. Additionally, TUS can be performed at the point of care, making it a more practical diagnostic tool in diverse clinical settings.

We conducted a systematic review and meta-analysis of the relevant studies on the use of TUS for diagnosing pneumonia in children, aiming to evaluate its diagnostic accuracy as an alternative method.

## 2. Methods

### 2.1. Search Strategy

The systematic review was conducted in accordance with the Preferred Reporting Items for Systematic Reviews and Meta-analyses (PRISMA) statement [9]. The protocol was registered in the international prospective register for systematic reviews, PROSPERO (CRD42023470677). We searched in Embase, PubMed, and Web of Science databases to identify studies that investigated the accuracy and role of TUS for diagnosing pneumonia in children up until July 2023 (see Appendix A for the details of the search strategy). The authors (Z.D. and C.S.) systematically searched all the titles and abstracts published in English that were relevant to our study and downloaded the full texts for further evaluation. At the same time, we also manually retrieved the reference lists of the relevant systematic reviews to include other relevant articles that were not captured in the systematic search to ensure the comprehensiveness of the literature search.

### 2.2. Selection Criteria

The articles included in the meta-analysis had to satisfy the following criteria: (1) patients aged below 16 years, (2) presenting symptoms and signs suggestive of pneumonia (including but not limited to fever, tachypnea or dyspnea, cough, diminished breath sounds, and rales), (3) exhibiting positive findings on thoracic ultrasound (such as lung consolidation, air bronchogram sign, pleural effusion, abnormal pleural line, and B-lines etc. The details can be found in Table 1), and (4) clinical diagnoses along with laboratory and chest imaging examinations were used as reference.

The following exclusion criteria were applied: (1) patients with severe respiratory diseases other than pneumonia, (2) patients with congenital anomalies or chronic lung diseases (their chest radiography or ultrasound findings may have other signs that interfere with the diagnosis of TUS and CXR due to the impact of the congenital anomalies or chronic diseases.), (3) patients over 16 years of age, (4) patients who had not undertaken TUS as a diagnostic tool, (5) case reports and studies without original data (such as reviews, comments, etc.), and (6) studies with incomplete data (did not report the sensitivity and specificity as the results or did not provide enough information to calculate the sensitivity and specificity) or where statistical effect size pooling was not feasible.

### 2.3. Data Extraction and Quality Assessment

Two authors (Z.D. and J.T.) independently extracted the following data from each included article: (1) the first author’s name, (2) the year and region of publication, (3) the study design and blinding, (4) the patient characteristics and sources, (5) the experience of the TUS operators, (6) the type of TUS probe, (7) the TUS diagnostic criteria for pneumonia along with the reference test and (8) the quantitative data required for statistical analysis. The data extracted by the above two authors was then compared by another author (C.S.), and disagreements were resolved by voting after a dialogue between the two members of the research team (C.S., and B.W.).

All the included studies were independently assessed for methodological quality by two reviewers (Z.D. and J.T.) using the Quality Assessment of Diagnostic Accuracy Studies-2 (QUADAS-2), a tool that evaluates the risk of bias and applicability across four domains of the included studies [10]. Any discrepancies between the two reviewers were discussed, and if a consensus could not be reached, they were resolved by involving the third author (C.S.).

### 2.4. Data Analysis

The Stata MP 14.0 software (National Center for Supercomputer Applications, Urbana, IL, USA) was used to analyze the diagnostic accuracy. The pooled sensitivity, specificity, positive likelihood ratio (PLR), negative likelihood ratio (NLR), and 95% confidence intervals (95% CI) were calculated based on the collected data for the sensitivity, specificity, true positive (TP), false positive (FP), false negative (FN), and true negative (TN). Forest plots and summary receiver operating characteristic curves (sROC) were generated. The Cochran Q-statistic and inconsistency (I^2^) were utilized to assess the heterogeneity while a meta-regression analysis was employed to explore the potential sources of heterogeneity. Random effects models were used for all the quantitative analyses.

## 3. Results

### 3.1. Study Selection

The database search, encompassing articles published up to July 2023, revealed a total of 252 articles that satisfied the study topic and selected keywords. Among these, 61 full-text articles were identified for further assessment and screening. Following the predefined inclusion and exclusion criteria, a final selection of 26 [11,12,13,14,15,16,17,18,19,20,21,22,23,24,25,26,27,28,29,30,31,32,33,34,35,36] observational studies (22 prospective and four retrospective) were included for the meta-analysis (Figure 1).

### 3.2. Study Characteristics

A total of 26 studies involving 4436 participants (2402 males and 2034 females; 54.1% vs. 45.9%) were included in the analysis. Among them, 2866 patients were diagnosed with pneumonia, and the remaining 1570 participants were not diagnosed with pneumonia. The diagnostic criteria for pneumonia included clinically relevant symptoms and evidence of imaging and/or laboratory tests. The specific details of the diagnostic reference tests for each study are presented in Table 2. The detailed characteristics of these studies are also presented in Table 2. Two studies were conducted in North America [14,29], sixteen in Europe [11,12,13,15,16,18,19,20,21,22,23,24,26,27,28,35], seven in Asia [17,25,30,32,33,34,36], and one in Africa [31]. The study subjects were mainly infants, children, and adolescents, except for two studies that included newborns [17,32]. A total of 23 studies involved pediatric patients admitted to either the wards or emergency departments, while three studies [15,17,32] involved patients in ICU (two studies in NICU and one in PICU). To avoid significant changes in the consolidation process due to the time passed between the two imaging tests, nearly all the patients included in our study (except Zhan et al. [22]) performed chest imaging and TUS consecutively within 24 h of hospitalization. Table 3 illustrates the details of the chest ultrasound used to diagnose pneumonia, including the operator, type of probe, blinding, and the diagnostic criteria of TUS. In addition, Table 4 describes the original data required for meta-analysis, including the TP, FP, TN, FN as well as the sensitivity and specificity.

### 3.3. Quality Assessment

QUADAS-2 was used to evaluate the quality of the included literature, and in general, most of the articles included in this study were of high quality, as elaborated Table 5. The potential for bias and limitations in terms of the applicability was negligible in the following domains: a precise definition of the target population and explicit inclusion and exclusion criteria for the study enrollment, a meticulous description of how the index and reference tests were employed to diagnose pneumonia, as well as a comprehensive presentation of the study design and methodology.

On the whole, the majority of the findings used in our study were deemed to be of good quality. A total of 24 trials (92.3%) included patients who underwent a CXR as part of their standard clinical procedure, while only two studies (7.7%) did not mention the utilization of CXRs as the controls. The included studies all incorporated specific criteria for inclusion and exclusion, along with details regarding the diagnosis of TUS. However, some studies involved patients from the ICU or neonates [15,17,32], and may not have objectively reflected the disease characteristics of the majority of the pediatric patients, thus inducing a potential risk of selection bias. There were two studies that did not mention whether the interpreters were blinded to the results of TUS [11,20], thus posing a risk of bias. In addition, the sex ratio of the included subjects was 54.1% to 45.9% (2402 males and 2034 females), which may cause bias due to the difference in the number of genders.

### 3.4. Details of TUS for Diagnosing Pediatric Pneumonia

In the analyzed articles, the level of experience among the ultrasound operators and interpreters varied greatly, ranging from novice radiologists or even pediatric residents who received only brief training in TUS for diagnostic purposes [15,22] to senior sonographers with extensive imaging expertise [24,28,33,34]. At present, most of the commonly used ultrasonic probes are 5–10 MHz high-frequency linear transducers and micro-convex transducers. The method of TUS examination proposed by Copetti and Cattarossi is currently the most widely referenced, enhanced, and applied approach in clinical research [11]. The TUS examination was employed to systematically scan the bilateral lung fields, dividing each hemi-thorax into three distinct regions to cover the entire lung: the anterior region delimited by the parasternal line and the anterior axillary line, the lateral region delimited by the axillary line, and finally, the posterior region delimited by the paravertebral line and the posterior axillary line. Within each of these regions, it was recommended to perform scans along the anatomical lines, including the parasternal, mid-clavicle, anterior axilla, central axilla, posterior axilla, central scapular, and paravertebral areas [11,15]. The probe was typically positioned vertically, diagonally, and parallel to the ribs of the anterior, lateral, and posterior thoraxes, respectively, for a comprehensive examination of each intercostal space [37,38].

### 3.5. Meta-Analysis

A total of 26 studies were included to assess the diagnostic accuracy of TUS in pediatric pneumonia. The statistical software was utilized to calculate the pooled sensitivity, specificity, positive likelihood ratio (PLR), negative likelihood ratio (NLR), and their corresponding 95% CI. Additionally, the forest plot was generated. A cutoff value of *p* < 0.05 was used to determine inconsistency.

For the diagnosis of pneumonia in children, TUS demonstrated a pooled sensitivity and specificity of 0.95 (95% CI, 0.92–0.97) and 0.94 (95% CI, 0.88–0.97), respectively (Figure 2). However, the I^2^ was 96.17% and 92.35%, respectively, showing significant heterogeneity. The PLR and NLR were 14.88 (95% CI, 7.92–27.95; Cochran Q-statistic = 417.52; *p* = 0.00) and 0.05 (95% CI, 0.03–0.09; Cochran Q-statistic = 772.79; *p* = 0.00), which also indicated significant heterogeneity in the data analysis (Figure 3). The area under the sROC curve (AUC) was determined to be 0.98 (95% CI, 0.97–0.99), indicating an excellent diagnostic accuracy with a slope towards the upper left, as depicted in Figure 4.

### 3.6. Publication Bias

Deek’s funnel plot asymmetry test was conducted to investigate potential publication bias, and the results indicated no significant evidence of publication bias among the included studies (*p* = 0.13). Therefore, we did not perform the trim and fill analysis.

### 3.7. Subgroup Analysis

We conducted a subgroup analysis from the following aspects in an attempt to provide a more detailed reflection of the reliability of TUS for diagnosing pneumonia in children. The findings are presented in Table 6.

#### 3.7.1. Patients Setting

Among the 26 studies, 15 studies conducted TUS in pediatric emergency departments. Of the remaining 11 studies, eight were conducted in the pediatric ward, two in the NICU, and one in the PICU. The subgroup analysis revealed that the sensitivity and specificity of the pediatric emergency departments were 0.93 (95% CI, 0.88–0.97) and 0.91 (95% CI, 0.84–0.95), respectively. The PLR was calculated as 10.05 (95% CI, 5.53–18.27), with a NLR of 0.07 (95% CI, 0.04–0.14). The AUC was determined to be 0.97 (95% CI, 0.95–0.98). On the other hand, for TUS performed in pediatric wards were found to be at a higher level of accuracy with sensitivity and specificity values of 0.95 (95% CI, 0.91–0.97) and 0.96 (95% CI, 0.82–0.99), respectively. The PLR was 22.81 (95% CI, 4.84–107.45), the NLR was 0.06 (95% CI, 0.03–0.09), and the AUC remained high at a value of 0.98 (95% Cl, 0.96–0.99). Due to the limited number of studies conducted in the NICU and PICU, the subgroup analysis was not available.

#### 3.7.2. Operator Experience

In six studies, the operators performing TUS had limited experience or received only a brief training (less than 1 month of ultrasound training or experience). The subgroup analyses revealed the following results. The sensitivity was 0.83 (95% CI, 0.67–0.93), with a specificity of 0.91 (95% CI, 0.88–0.94). The PLR was 9.78 (95% CI, 6.69–14.30), and the NLR was 0.18 (95% CI, 0.08–0.39). The AUC was calculated as 0.93 (95% CI, 0.90–0.95). TUS for the remaining 20 studies were performed and interpreted by experienced sonographers (more than 3 years experience in ultrasonic clinical diagnosis) or highly trained radiologists and pediatricians (more than 5 years of clinical experience and more than 1 year of ultrasonic diagnosis training). The subgroup analysis yielded the following results: a sensitivity of 0.96 (95% CI, 0.94–0.97), a specificity of 0.94 (95% CI, 0.87–0.98), a PLR of 16.87 (95% CI, 7.06–40.29), an NLR of 0.04, (95% CI, 0.03–0.06), and an AUC of 0.98 (95% CI, 0.97–0.99).

#### 3.7.3. Probe Selection

Out of all the 26 studies, 13 selected linear and convex probes for the TUS diagnosis, 11 selected linear probes, and the remaining two studies used convex probes. The subgroup analysis revealed that the sensitivity and specificity of the diagnoses using the linear combined convex probes were 0.95 (95% CI, 0.92–0.97) and 0.95 (95% CI, 0.87–0.98), respectively. The PLR was calculated as 18.23 (95% CI, 6.89–48.19), with a NLR of 0.05 (95% CI, 0.03–0.09). The AUC was determined to be 0.99 (95% CI, 0.97–0.99). The results for the subgroups of the studies using only the linear probe were as follows: sensitivity, 0.94 (95% CI, 0.87–0.97); specificity, 0.94 (95% CI, 0.85–0.98); PLR, 14.77 (95% CI, 5.85–37.28); NLR, 0.06 (95% CI, 0.03–0.15); AUC, 0.98 (95% CI, 0.96–0.99).

According to the results of the subgroup analysis, the setting of the patients and probe selection did not significantly affect the diagnostic accuracy of TUS in pediatric pneumonia. However, whether the operators had a good experience in performing TUS had a certain effect on the diagnostic accuracy. The performance of the experienced sonographers in the diagnosis of pneumonia in children was better than that of the inexperienced sonographers.

## 4. Discussion

### 4.1. Pediatric Pneumonia and Its Current Status of Diagnosis

Pneumonia is the single leading infectious cause of pediatric hospitalization and death worldwide. Pneumonia caused an estimated 740,000 children’s death under the age of 5 years in 2019 alone, accounting for 14% of all infant deaths under the age of five and 22% of all deaths among children aged 1–5 years globally [39,40]. With the improvement of medical services and living standards, and the global attention to the prevention and treatment of pneumonia in children, the morbidity and mortality of pneumonia in children have presented a downward trend [41]. Although the number of pneumonia cases and mortality in children have decreased significantly, the number of hospital visits and hospitalization rates of pneumonia patients have increased in developing countries and some developed countries [41]. In addition, the COVID-19 pandemic has had a significant impact on the epidemiology of related respiratory diseases in recent years, showing a different epidemiology of pneumonia from that before the pandemic [42,43]. Therefore, in the current situation, the early detection, diagnosis, and treatment of pneumonia in children has become essential.

Currently, the diagnosis of childhood pneumonia still poses many problems, particularly in resource-limited settings. The difficulties in its diagnosis can be attributed to several factors. Firstly, the symptoms and signs of pneumonia are often inconspicuous in infected children due to their age, physical conditions, complications, and etiology of infection, thus lacking specificity and complicating the diagnostic process [44,45]. Secondly, due to the inherent characteristics of pediatric patients, such as their limited ability to articulate their medical history and symptoms accurately, as well as their reduced tolerance towards illness, clinicians face challenges in making informed judgments. Currently, the prevailing approach for diagnosing pneumonia in children relies on CXR. In cases where CXR fails to provide a definitive diagnosis for pneumonia, further evaluation with CT is warranted [46]. However, these imaging modalities are not suitable for pediatric patients. First of all, CXR, which is commonly used as the initial diagnostic tool, has limited sensitivity (86.2%, refer to the data in Balk et al.) when used alone and requires children to be exposed to radiation. Additionally, there is some inter-observer variability in its interpretation [47]. Secondly, although CT offers a high diagnostic accuracy, it involves excessive radiation exposure, high costs, and sometimes necessitates the sedation of children [48]. Consequently, their application in the diagnosis of common pediatric pneumonia is restricted.

### 4.2. The Feasibility and Performance of TUS in the Diagnosis of Children

In the past decade, significant advancements in clinical diagnostic technology have propelled ultrasound technology to achieve remarkable progress. The resolution of ultrasonic instruments has undergone substantial improvements, thereby significantly enhancing the detection rate of abnormal disease signs. The convenience of ultrasound equipment has also been enhanced. From larger instruments that were initially fixed only in the consulting room, we now have portable bedside ultrasound machines and even pocket (hand-held) ultrasound devices that can be easily carried around. With the continuous development of modern science and technology, ultrasound diagnosis has transcended conventional boundaries and is no longer confined solely to diagnosing lesions in parenchymatous organs. Nowadays, the application of ultrasound has progressively expanded in the diagnosis of thoracic, gastrointestinal, and other hollow organ diseases.

In recent years, an increasing number of researchers have turned their attention to the application of thoracic ultrasound for the diagnosis of chest diseases. Previous studies have confirmed that TUS exhibits a high sensitivity and specificity for detecting various pleural diseases in adults [49]. A meta-analysis investigating TUS for diagnosing community-acquired pneumonia (CAP) in adults revealed that TUS demonstrated a higher sensitivity and specificity, resulting in a superior diagnostic accuracy than CXR [50]. TUS can assist clinicians in diagnosing CAP among adult patients. However, the role of TUS in pediatric pneumonia diagnosis remains uncertain.

The feasibility of TUS for the diagnosis of pneumonia is mainly based on the anatomical relationship between the thorax and lung. The superficial layer of the thoracic cavity is composed of subcutaneous tissue and muscle, and the deep layer is composed of pleural tissue attached to the inner surface of the thoracic cavity and the surface of the lungs, which are air-filled organs. Thus, the large variation in acoustic impedance at the pleura and lung junction results in the generation of horizontal artifacts, which are seen as a series of equidistant and parallel echo lines under the pleura, known by sonographers as A-lines. The vertical comet tail-like artifacts produced by the pleural line are called B-lines, which are not present in the normal lung. The B line originates from the pleural line, exhibits a clear demarcation, extends posteriorly to replace the A-line at the margin, and demonstrates movement corresponding to lung sliding [51,52]. Chest ultrasonography commonly relies on the aforementioned artifact analysis. The ultrasound findings for pneumonia primarily manifest as hypoechoic consolidation areas with varying sizes and shapes, characterized by irregular borders [33]. The echotexture may exhibit homogeneity or heterogeneity [53], while lung sliding may be diminished or even absent [7].

The chest wall structure in children is characterized by its thinness, smaller thoracic width, and anteroposterior diameter, which renders it more conducive for obtaining high-quality images using TUS imaging. Furthermore, the greatest advantage of ultrasound diagnosis in pediatric cases lies in its ability to eliminate radiation exposure, thereby significantly reducing the potential medical risks associated with medical treatment. Moreover, due to the convenience of imaging equipment and the low requirements for usage conditions, ultrasound examinations can be conveniently performed at the point of care without transportation, which is especially beneficial for bedside diagnoses in ICU and emergency department settings or for pediatric pneumonia screening in remote areas. In conclusion, TUS exhibits numerous advantages and holds great potential for diagnosing pneumonia in children.

Various ultrasound transducers are used for conducting chest ultrasound examinations. Contemporary ultrasound devices typically incorporate three standard probes: linear, curvilinear, and phased arrays, each of which plays a crucial role in TUS. Some providers might possess a micro-convex transducer that is specifically designed for TUS, but this is less common. A high-frequency (8–12 MHz) linear probe is advantageous for delineating the anterior pleural line with precision, despite its limited depth penetration. Conversely, a low-frequency curvilinear probe is instrumental for visualizing the dependent lung regions. The phased array probe (2.5–5 MHz), with its reduced footprint and intermediate frequency, offers an optimal balance. However, it poses challenges in accurately delineating the pleural line with this transducer in patients [51]. The operator typically selects probes of different types based on the specific location of the lesion and the physical characteristics of the patient. Most of the studies included in this meta-analysis selected linear and convex probes with different frequencies. A subgroup analysis of the probe selection revealed a minimal variation in the diagnostic performance of TUS when different probes were used, which was potentially attributed to the limited number of included studies and predominantly involved pediatric patients with pneumonia, where the lesion was typically localized in a relatively fixed position. Further research is needed in the future to explore the possible impact of the use of different probes for TUS diagnosis.

After a rigorous and systematic literature review and screening, a total of 26 original studies were included in this meta-analysis to evaluate the performance of TUS in the diagnosis of pneumonia in children. After the statistical analysis, our findings revealed that TUS exhibited a high sensitivity and specificity, with a pooled sensitivity of 0.95 (95% CI, 0.92–0.97) and a pooled specificity of 0.94 (95% CI, 0.88–0.97), surpassing the previous meta-analysis conducted by Lu et al. [54]. On the other hand, the combined PLR and NLR were 14.88 (95% CI, 7.92–27.95) and 0.05 (95% CI, 0.03–0.09), respectively, and the AUC was 0.98 (95% CI, 0.97 = 0.99), both indicating the exceptional diagnostic performance of TUS in pediatric pneumonia detection. Of all the 26 included studies, one (Zhan et al.) showed a poor sensitivity (40.0%, 95% CI, 30.0–51.0%) to TUS, and we analyzed the potential confounders that could have contributed to this result. First, to eliminate any bias, the operator (a pediatric resident with minimal practical ultrasound experience) in this study was blinded to all the information regarding the patient, including the physical examination, results of the blood tests and chest radiography fundings, while other studies were only blinded to the chest radiography fundings. Second, because of unforeseen events, they were forced to change ultrasound equipment after the first 24 patients. This made it difficult to assess whether the observed significant difference in the sensitivity was caused by the equipment change or for other reasons. Finally, due to practical challenges, the LUS and CR were not always performed consecutively in this study, which may have potentially exerted a great impact on the diagnostic accuracy of TUS.

However, there was considerable heterogeneity in the sensitivity and specificity among the 26 studies (I^2^ > 90%). We hypothesized that the covariates of the “ultrasound operator experience”, “different ultrasound equipment”, “type of wards”, and “choice of reference tests” might have been potential factors contributing to the heterogeneity. The discussion of the diagnostic heterogeneity of TUS in the current literature was replicated and corroborated in this analysis, which was similar to the results of the analysis by Balk et al. [55]. The results of the subgroup analysis showed that experienced operators used TUS to diagnose pneumonia with higher accuracy, which was consistent with the results predicted in advance. Nevertheless, the non-imaging clinicians with systematic training also had a better sensitivity and specificity in the diagnosis of TUS. For instance, in a study conducted by Esposito et al. [15], pediatric residents who received only 7 h of training demonstrated remarkable sensitivity (98%) and specificity (95%) when utilizing TUS for the diagnosis of pediatric pneumonia. The International Liaison Committee on Lung Ultrasound also indicated that TUS is a fundamental ultrasound examination method that is easily acquired and applied [56]. It can be seen that the learning cost of TUS is relatively low, making it accessible even for beginners. Moreover, TUS exhibits a consistent diagnostic accuracy, rendering it suitable for widespread implementation and adoption in clinical settings.

### 4.3. Limitations and Future Prospects

Some limitations of our study should be acknowledged. Firstly, there was a dearth of standardized procedures for testing the research validity and assessing the quality requirements. Consequently, we did not conduct an analysis on the internal consistency of each sample or investigate its impact on the overall success rate of the TUS diagnosis. Secondly, due to the absence of corresponding data in some of the included literature, our meta-analysis lacked controls. We solely collected and analyzed the data pertaining to the pneumonia diagnosis by TUS without examining and comparing the accuracy of CXR or other diagnostic methods. Thus, we were unable to quantitatively demonstrate the improvement in the TUS accuracy compared to the other diagnostic approaches. In addition, due to ethical constraints, our included studies were unable to employ traditional CT as the diagnostic reference standard. Instead, we utilized CXR alone or in combination with clinical diagnosis and laboratory tests as the reference criteria. However, given the limited number of included studies, we were unable to establish this combined test as a criterion. Therefore, it was important to interpret our findings by taking that into account. Finally, the majority of the studies we included were single-center and small in scale. Therefore, it is our expectation that future researchers will conduct more rigorous evaluations using larger sample sizes and multicenter clinical studies.

Currently, the differentiation of pneumonia pathology is limited to clinical manifestations and laboratory tests. Although there is increasing evidence suggesting that TUS is more accurate than CXR for diagnosing pneumonia in children [47], the literature on the differential etiology of pneumonia remains scarce. This requires further analysis of current ultrasound findings in the gray zone, encompassing abnormalities in the pleural line, sub-centimeter solid nodules, and focal confluent B-lines. Additionally, there is a lack of corresponding research to substantiate the optimal timing for TUS diagnosis. Previous studies have conducted follow-ups on patients undergoing TUS and observed that as clinical interventions are implemented and the condition improves, the lung consolidation shadow gradually diminishes in the ultrasound images. This phenomenon may potentially result in a decline in the diagnostic sensitivity of TUS. Therefore, future studies are needed to further explore the optimal window time of TUS for the diagnosis of pneumonia to achieve more standardized protocols for clinical application. The results of this subgroup analysis revealed the influence of varying levels of TUS expertise on the accuracy of pneumonia diagnosis, highlighting the need to develop a standardized TUS training curriculum and quantify the duration required for beginners to attain proficiency in performing TUS and interpreting the imaging results accurately. Therefore, TUS can be more effectively promoted and popularized as a routine diagnostic technique in the workflow of clinicians [57].

Recently, significant advancements in the application of artificial intelligence (AI) across various medical domains have catalyzed the development and innovation of novel AI-based diagnostic technologies [58]. Deep learning is an advanced stage of artificial intelligence, which uses multilayer neuronal cascade learning to hierarchically extract raw data, thus simulating human neural networks [59,60]. Previous studies have reported the application of deep learning methods in the diagnosis of pneumonia in children [61,62]. It can be seen that AI technology has great prospects in the promotion of TUS, especially in economically underdeveloped areas or primary medical institutions. By employing an automated intelligent diagnosis system for TUS, the detection rate of pneumonia in children can be enhanced while simultaneously improving the efficiency of clinicians.

## 5. Conclusions

In conclusion, TUS is a highly promising imaging diagnostic technique that can serve as a valuable adjunct to CXR and physical examination. The most notable advantage of this approach for pediatric patients lies in its ability to avoid ionizing radiation exposure. Our meta-analysis results demonstrated that TUS exhibits excellent sensitivity and specificity for diagnosing pneumonia among children, with the added benefit of being suitable even for novices inexperienced with ultrasound interpretation. Therefore, actively promoting the widespread adoption of this convenient diagnostic technology among general pediatricians can yield significant benefits for both medical practitioners and patients across diverse clinical settings. However, it is important to acknowledge that this conclusion was solely based on the clinical studies that met our specific inclusion criteria, necessitating future large-scale, multicenter trials that are well designed to validate and evaluate the conclusions.

## Figures and Tables

**Figure 1 diagnostics-13-03457-f001:**
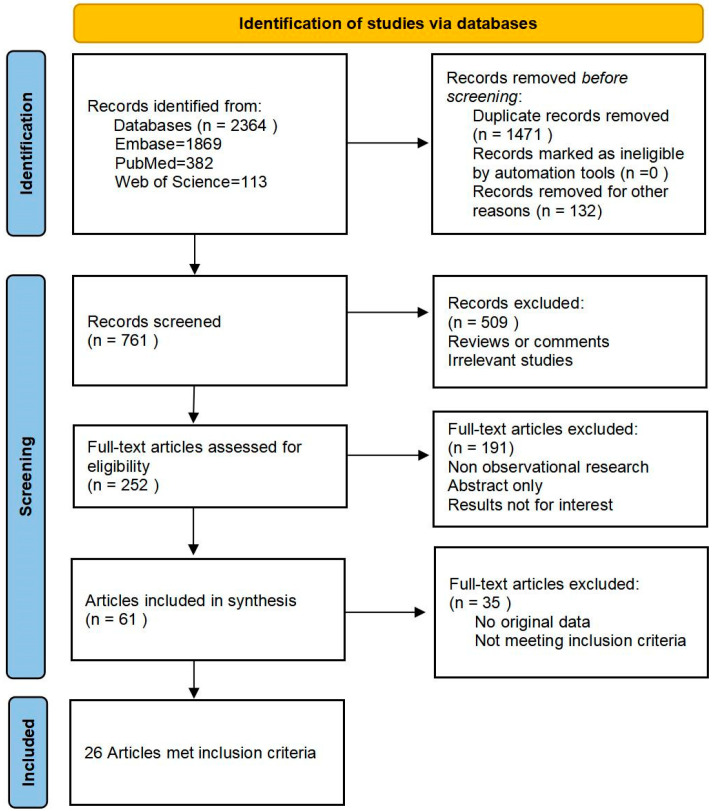
PRISMA flowchart of the literature search process.

**Figure 2 diagnostics-13-03457-f002:**
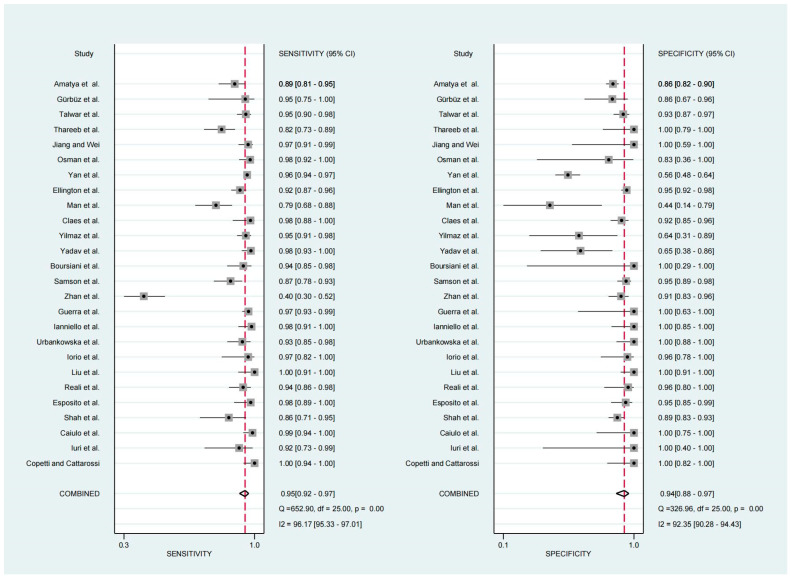
Forest plot depicting the sensitivity and specificity [11,12,13,14,15,16,17,18,19,20,21,22,23,24,25,26,27,28,29,30,31,32,33,34,35,36].

**Figure 3 diagnostics-13-03457-f003:**
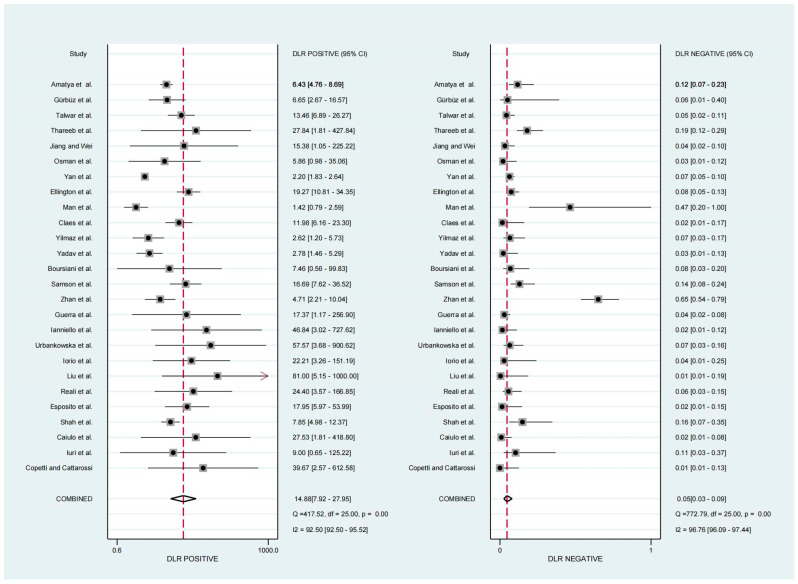
Forest plot depicting the PLR and NLR [11,12,13,14,15,16,17,18,19,20,21,22,23,24,25,26,27,28,29,30,31,32,33,34,35,36].

**Figure 4 diagnostics-13-03457-f004:**
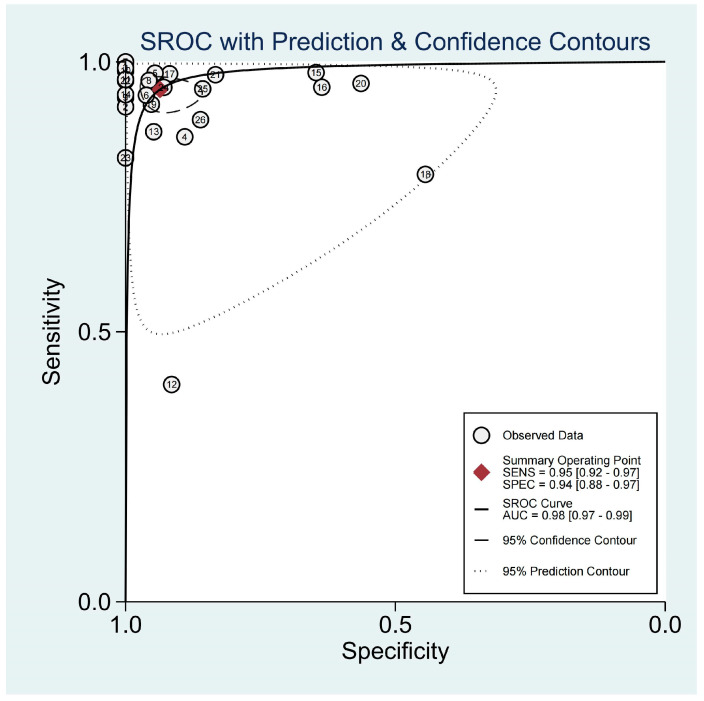
sROC graph of the TUS diagnostic performance. (The numbers in the figure were represented the 26 included studies.)

**Table 1 diagnostics-13-03457-t001:** Definition and meaning of the TUS fundings.

TUS fundings	Definition/Meaning
Lung sliding	Rhythmic movement of the pleural line with respiration (a normal funding).
Lung consolidation	Hypoechogenic area with irregular margins and adjacent comet tail artifacts along a non-homogeneous echo texture. Liver-like (“hepatized”) area with fluid and air bronchograms (static or dynamic).
Pleural effusion	Anechoic space between the visceral and parietal pleura.
Air bronchogram	Punctate or branching hyperechoic images reflecting the airways made visible by the surrounding fluid/inflammation.
Fluid bronchogram	Identified as anechoic tublar structures, with hyperechoic walls. It may indicate obstructive pneumonia.
A-lines	Horizontal lines equally spaced from the pleural line, representing artifacts generated by subpleural air (a normal finding).
B-lines	Hyperechoic artifact images erasing the A-line, extending from the pleural line to the end of the screen, emerging with lung movements and extending in a vertical manner, and classified as focal, multiple, or confluent. The presence of these artifacts can be attributed to fluid-rich subpleural interlobular septae, which are surrounded by air and associated with various pathological findings.

**Table 2 diagnostics-13-03457-t002:** Characteristics of the involved studies.

Authors	Year	Country	Study Type (Observational)	Patients Setting	Reference Standard	Sample Size	Sex (Male/Female)	Ages
Copetti and Cattarossi [11]	2008	Italy	Prospective	Pediatric ED	Clinical diagnosis + CXR	79	37/42	5.1 ± 5.0 years
Iuri et al. [12]	2009	Italy	Prospective	Pediatric ED	CXR	28	17/11	4.5 ± 4.9 years
Caiulo et al. [13]	2013	Italy	Prospective	Pediatric ward	Clinical diagnosis + CXR + blood results	102	53/49	5.0 ± 3.0 years
Shah et al. [14]	2013	USA	Prospective	Pediatric ED	CXR	200	112/88	2.6 ± 6.2 years
Esposito et al. [15]	2014	Italy	Prospective	PICU	CXR	103	56/47	5.6 ± 4.6 years
Reali et al. [16]	2014	Italy	Prospective	Pediatric ward	Clinical diagnosis + CXR + blood results	107	61/46	4.0 ± 3.0 years
Liu et al. [17]	2014	China	Prospective	NICU	Clinical diagnosis + CXR + blood results	80	43/37	NR
Iorio et al. [18]	2015	Italy	Retrospective	Pediatric ward	Clinical diagnosis + CXR + laboratory findings	52	27/25	3.5 ± 3.1 years
Urbankowska et al. [19]	2015	Poland	Prospective	Pediatric ward	Clinical diagnosis + CXR + laboratory findings	106	69/37	Median = 52.5 months
Ianniello et al. [20]	2016	Italy	Retropective	Pediatric ED	Clinical diagnosis + CXR	84	44/40	Mean = 6.0 years
Guerra et al. [21]	2016	Italy	Prospective	Pediatric ED	Clinical diagnosis + CXR + blood results	222	108/114	3 months–16 years
Zhan et al. [22]	2016	Denmark	Prospective	Pediatric ED	CXR + blood results + microbiological testing	164	93/71	Median = 1.5 years
Samson et al. [23]	2016	Spain	Prospective	Pediatric ED	Clinical diagnosis + CXR	200	116/84	Median = 29.5 months
Boursiani et al. [24]	2017	Greece	Prospective	Pediatric ED	Clinical diagnosis + CXR	69	27/42	Median = 4.5 years
Yadav et al. [25]	2017	India	Prospective	Pediatric ED	Clinical diagnosis + CXR	118	65/53	26.2 ± 19.6 months
Yilmaz et al. [26]	2017	Turkey	Prospective	Pediatric ED	Clinical diagnosis + CXR	160	88/72	3.3 ± 4.0 years
Claes et al. [27]	2017	Belgium	Prospective	Pediatric ED	Clinical diagnosis + CXR	143	77/66	Mean = 3.4 years
Man et al. [28]	2017	Romania	Retrospective	Pediatric ED	Clinical diagnosis + laboratory findings	81	42/39	6.5 ± 4.7 years
Ellington et al. [29]	2017	USA	Prospective	Pediatric ED/ward	Clinical diagnosis + CXR + laboratory findings	421	258/163	20.0 ± 14.6 months
Yan et al. [30]	2020	China	Retrospective	Pediatric ward	Clinical diagnosis + CXR + CT	925	435/490	12.5 ± 3.1 years
Osman et al. [31]	2020	Egypt	Prospective	Pediatric ward	Clinical diagnosis + CXR	90	36/54	Median = 2.2 years
Jiang and Wei [32]	2022	China	Prospective	NICU	Clinical diagnosis + CXR + laboratory findings	122	64/58	NR
Thareeb et al. [33]	2022	Iraq	Prospective	Pediatric ward	Clinical diagnosis + CT	106	70/36	6.5 ± 4.1 years
Talwar et al. [34]	2022	India	Prospective	Pediatric ward	Clinical diagnosis	261	168/93	4.3 ± 4.4 years
Gürbüz et al. [35]	2023	Turkey	Prospective	Pediatric ED	Clinical diagnosis + CXR	48	27/21	5.4 ± 5.3 years
Amatya et al. [36]	2023	Nepal	Prospective	Pediatric ED	Clinical diagnosis + CXR	365	209/156	Median = 16.5 months

ED, emergency department; PICU, pediatric intensive care unit; NICU, neonatal intensive care unit; CXR, chest X-ray; CT, computed tomography, NR: not reported.

**Table 3 diagnostics-13-03457-t003:** Ultrasound details of the enrolled studies.

Study	Blinding *	Operator	LUS Probe	Diagnostic Criteria
Copetti and Cattarossi (2008) [11]	NM	Emergency physician	3.5–5 MHz convex probe and high-resolution 7.5–10 MHz linear probe	Lung consolidation, bronchograms, and pleural effusion
Iuri et al. (2009) [12]	Yes	Radiologist	Broadband (2–5 MHz) convex probe and high-frequency broadband (5–12 MHz) linear probe	Lung consolidation and pleural effusion
Caiulo et al. (2013) [13]	Yes	Pediatrician with ultrasound experience	High-resolution linear probe with frequencies ranging from 6–12 MHz	B-lines, consolidation, air pleural line abnormalities, and pleural effusion
Shah et al. (2013) [14]	Yes	Emergency physician with ultrasound experience	Linear array transducer at 7.5–10 MHz	Lung consolidation, air bronchogram, and B-lines
Esposito et al. (2014) [15]	Yes	Pediatric resident with limited ultrasound experience	Convex 2.5–6.6 MHz probe and linear 7.5–12 MHz probe	B-lines, interruption of pleural line, Consolidation, and pleural effusion
Reali et al. (2014) [16]	Yes	Pulmonologist	Linear probe (7.5–10 MHz)	B-lines, consolidation, air pleural line abnormalities, and pleural effusion
Liu et al. (2014) [17]	Yes	Experienced physician	High-frequency linear 9–12 MHz probe	Pleural line abnormalities, lung consolidation, disappearance of lung sliding, and lung pulse
Iorio et al. (2015) [18]	Yes	Pediatrician with ultrasound experience	5–10 MHz linear probe	Lung consolidation, bronchograms, superficial fluid alveologram, and pleural effusion
Urbankowska et al. (2015) [19]	Yes	Pediatric sonographer	Convex 3–7 MHz probe and linear 5–9 MHz probe	Lung consolidation, bronchograms, pleural effusion, and pleural line abnormalities
Ianniello et al. (2016) [20]	NM	Radiologist	Curved array 4 MHz multifrequency probe and linear 7.5–10 MHz probe	Lung consolidation, B-lines, air bronchogram, pleural line abnormalities, and pleural effusion
Guerra et al. (2016) [21]	Yes	Pediatrician with ultrasound experience	High-resolution 7.5–10-MHz linear probe ± 3.5–5-MHz convex probe	Lung consolidation, air and fluid bronchogram, pleural effusion, and B-lines
Zhan et al. (2016) [22]	Yes	Pediatric resident with limited ultrasound experience	Two linear probes (5–10 MHz and 5–13 MHz)	Subpleural consolidation, air bronchogram, and pleural effusion
Samson et al. (2016) [23]	Yes	Pediatricians and pediatrics with varying ultrasound experience	6–15 MHz linear probe	Lung consolidation, air bronchogram, and pleural effusion
Boursiani et al. (2017) [24]	Yes	Experienced pediatric radiologist	5–8 MHz micro-convex, 5–12 MHz linear array, and 3–5 MHz convex transducers)	B-lines, lung consolidation, air or fluid bronchogram, and pleural effusion
Yadav et al. (2017) [25]	Yes	Trained radiologist	high-resolution micro-convex transducer with the depth of 8 cm	Lung consolidation, air bronchogram, fluid alveologram, B-lines, and pleural effusion
Yilmaz et al. (2017) [26]	Yes	Experienced pediatric emergency physician	6–13 MHz linear probe	Consolidation, air and fluid bronchograms, B-lines, and pleural effusion
Claes et al. (2017) [27]	Yes	Pediatric radiologist	Linear 5–12 MHz probe ± convex 4–9 MHz probe	Lung consolidation
Man et al. (2017) [28]	Yes	Senior radiologist	Convex (7–11 MHz) and linear (3.5–5 MHz) probes	Consolidation, air bronchogram, and pleural effusion
Ellington et al. (2017) [29]	Yes	Trained general pracitioner	HFL38/13–6 MHz linear transducer.	Consolidation with a pleural effusion
Yan et al. (2020) [30]	Yes	Experienced sonographer	7.5 MHz linear probes	B-lines, consolidation, air bronchogram, and pleural effusion
Osman et al. (2020) [31]	Yes	Radiologist	3–5 MHz convex transducer and high-frequency 5–12-MHz linear array probe	B-lines, consolidation, air bronchogram, and pleural effusion
Jiang and Wei (2022) [32]	Yes	Senior radiologist with professional training	High-frequency linear array probe (frequency 10–12 MHz)	Lung consolidation, air bronchogram, pleural effusion, B-lines, and pleural line abnormalities
Thareeb et al. (2022) [33]	Yes	Senior radiologist with more than five years of experience	6–9 MHz linear probe and 3–5 MHz convex probe	Lung consolidation, B-lines, air or liquid bronchogram, and pleural effusion
Talwar et al. (2022) [34]	Yes	Expert clinicosonologist	Micro-convex transducer with curvilinear (3.5–5 MHz) and linear (7.5–10 MHz) probes	B-lines, consolidation, air bronchogram, fluid bronchogram, and pleural effusion
Gürbüz et al. (2023) [35]	Yes	Pediatric resident with ultrasound training	Linear 3–13 MHz and convex 2–6 MHz probes	Lung consolidation, subpleural consolidation, pleural effusion, and B-lines
Amatya et al. (2023) [36]	Yes	Expert sonographer	Curvilinear probe	B-lines and subpleural consolidation

NM, not mentioned. *: means the TUS operators had no contact with the radiologists and were blinded to the chest radiography findings.

**Table 4 diagnostics-13-03457-t004:** Individual and cumulative statistics for the included studies.

Study	Participants	True Positives	False Positives	True Negatives	False Negatives	Sensitivity (95% CI, %)	Specificity (95% CI, %)	Positive PV (%)	Negative PV (%)
Copetti and Cattarossi [11]	79	60	0	19	0	100% (94.0–100)	100% (82.4–100)	100%	100%
Iuri et al. [12]	28	22	0	4	2	91.7% (NA)	100% (NA)	100%	66.70%
Caiulo et al. [13]	102	88	0	13	1	98.9% (93.9–100)	100% (75.3–100)	100%	92.90%
Shah et al. [14]	200	31	18	146	5	86.1% (71.0–94.0)	89% (83.0–93.0)	63.3%	96.70%
Esposito et al. [15]	103	47	3	52	1	97.9% (88.9–99.9)	94.5% (84.9–98.9)	94.00%	98.10%
Reali et al. [16]	107	76	1	25	5	93.8% (86.2–98.0)	96.2% (80.4–99.9)	98.70%	83.30%
Liu et al. [17]	80	40	0	40	0	100% (NA)	100% (NA)	100%	100%
Iorio et al. [18]	52	28	1	22	1	96.6% (82.2–99.9)	95.7% (78.1–99.9)	96.60%	95.70%
Urbankowska et al. [19]	106	71	0	30	5	93.4% (NA)	100% (NA)	100%	85.70%
Ianniello et al. [20]	84	60	0	23	1	98.4% (91.2–100)	100% (85.2–100)	100%	95.80%
Guerra et al. [21]	222	207	0	8	7	96.7% (93.4–98.7)	100% (63.1–100)	100%	53.30%
Zhan et al. [22]	164	33	7	75	49	40.0% (30.0–51.0)	91% (83.0–96.0)	82.50%	60.50%
Samson et al. [23]	200	74	6	109	11	87.1% (78.0–93.4)	94.8% (89.0–98.1)	92.50%	90.80%
Boursiani et al. [24]	69	62	0	3	4	93.9% (NA)	100% (NA)	100%	42.90%
Yadav et al. [25]	118	99	6	11	2	98.0% (93.0–99.8)	64.7% (38.3–85.8)	94.30%	84.60%
Yilmaz et al. [26]	160	142	4	7	7	95.3% (90.6–98.1)	63.6% (30.8–89.1)	97.30%	50.00%
Claes et al. [27]	143	44	8	90	1	98% (88.0–100)	92% (84.0–96.0)	84.60%	98.90%
Man et al. [28]	81	57	5	4	15	79.2 (68.0–87.8)	44.4 (13.7–78.8)	91.90%	21.10%
Ellington et al. [29]	421	176	11	219	15	92.2% (NA)	95.2% (NA)	94.10%	93.60%
Yan et al. [30]	925	745	65	84	31	96.0% (NA)	56.4% (NA)	92.00%	73.00%
Osman et al. [31]	90	82	1	5	2	97.6% (91.7–99.7)	83.3% (35.9–99.6)	98.80%	71.40%
Jiang and Wei [32]	122	111	0	7	4	96.5% (NA)	100% (NA)	100%	63.60%
Thareeb et al. [33]	106	74	0	16	16	82.2% (NA)	100% (NA)	100%	50.00%
Talwar et al. [34]	261	141	8	105	7	95.3% (90.5–98.1)	92.9% (86.5–96.9)	94.60%	93.75%
Gürbüz et al. [35]	48	19	4	24	1	95% (75.1–99.9)	85.7% (67.3–95.8)	82.60%	96.00%
Amatya et al. [36]	365	75	39	242	9	89.3% (81.0–95.0)	86.1% (82.0–90.0)	65.80%	96.41%

PV, predictive value; NA, not available.

**Table 5 diagnostics-13-03457-t005:** Risk of bias and applicability judgments in QUADAS-2.

Study	Patient Selection	Applicability	Index Test	Applicability	Reference Standard	Applicability	Flow and Timing
Copetti and Cattarossi [11]	U	L	U	L	U	L	U
Iuri et al. [12]	L	L	L	L	U	L	L
Caiulo et al. [13]	U	U	L	L	L	L	L
Shah et al. [14]	H	L	L	L	L	L	L
Esposito et al. [15]	H	L	L	L	L	L	L
Reali et al. [16]	L	L	L	L	L	L	L
Liu et al. [17]	H	L	L	L	L	L	L
Iorio et al. [18]	H	L	L	L	L	L	U
Urbankowska et al. [19]	L	L	L	L	L	L	L
Ianniello et al. [20]	H	L	L	L	U	L	L
Guerra et al. [21]	L	L	L	L	L	L	L
Zhan et al. [22]	L	L	L	L	L	L	H
Samson et al. [23]	L	L	L	L	L	L	L
Boursiani et al. [24]	L	L	L	L	L	L	L
Yadav et al. [25]	L	L	L	L	L	L	U
Yilmaz et al. [26]	U	L	U	L	L	L	U
Claes et al. [27]	L	L	L	L	L	L	L
Man et al. [28]	L	L	L	L	L	L	L
Ellington et al. [29]	L	L	L	L	L	L	L
Yan et al. [30]	L	L	L	L	L	L	U
Osman et al. [31]	L	L	L	L	H	L	L
Jiang and Wei [32]	H	L	U	L	L	L	U
Thareeb et al. [33]	L	L	U	L	U	L	L
Talwar et al. [34]	L	L	L	L	L	L	L
Gürbüz et al. [35]	U	L	L	L	L	L	U
Amatya et al. [36]	L	L	U	L	L	L	L

H, high; L, low; U, unclear.

**Table 6 diagnostics-13-03457-t006:** Subgroup analysis of the included studies by the different study characteristics.

Variable	*N*	Sensitivity (95% CI)	Specifcity (95% CI)	Positive LR	Negative LR	AUC
Patients settings						
ED	15	0.93 (0.88–0.97)	0.91 (0.84–0.95)	10.05 (5.53–18.27)	0.07 (0.04–0.14)	0.97 (0.95–0.98)
Ward	8	0.95 (0.91–0.97)	0.96 (0.82–0.99)	22.81 (4.84–107.45)	0.06 (0.03–0.09)	0.98 (0.96–0.99)
NICU *	2	NA	NA	NA	NA	NA
PICU *	1	NA	NA	NA	NA	NA
Operator experience						
Limited	6	0.83 (0.67–0.93)	0.91 (0.88–0.94)	9.78 (6.69–14.30)	0.18 (0.08–0.39)	0.93 (0.90–0.95)
Experienced	20	0.96 (0.94–0.97)	0.94 (0.87–0.98)	16.87 (7.06–40.29)	0.04 (0.03–0.06)	0.98 (0.97–0.99)
Probe selection						
Linear	11	0.94 (0.87–0.97)	0.94 (0.85–0.98)	14.77 (5.85–37.28)	0.06 (0.03–0.15)	0.98 (0.96–0.99)
Convex *	2	NA	NA	NA	NA	NA
Linear + convex	13	0.95 (0.92–0.97)	0.95 (0.87–0.98)	18.23 (6.89–48.19)	0.05 (0.03–0.09)	0.99 (0.97–0.99)

*N*, number of studies; LR, likelihood ratio; AUC, area under the sROC curve; NA, not available. *: The subgroup analysis was not available due to quantitative limitations.

## Data Availability

No new data were created or analyzed in this study. Data sharing is not applicable to this article.

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
