# Peer review of "Accuracy of Thoracic Ultrasonography for the Diagnosis of Pediatric Pneumonia: A Systematic Review and Meta-Analysis"

_diagnostics, 2023, doi:10.3390/diagnostics13223457_

Round 1
Reviewer 1 Report
Comments and Suggestions for Authors
Overall:
This is a manuscript evaluating the sensitivity and specificity of lung ultrasound for pneumonia evaluation in a wide range of pediatric patients through a meta-analysis of selected 26 published studies, consisting of 4436 patients, and assessing variability based on U/S location and training. Generally, the manuscript is not novel, as previous meta-analysis of lung U/S has been done before, and in more detail. In order to improve the caliber and strength of this manuscript, more detail will be needed.
Major Comments:
1. Line 74 – why were patients with chronic lung diseases excluded? They are especially prone to getting pneumonia. This needs to be explained. Also, the NICU studies that were included, how does the author know if those did not have chronic lung disease of prematurity?
2. Line 78 – what constitutes incomplete data?
3. Line 108 – explain what PRISMA guideline is, and why this was used
4. Line 110 – both prospective and retrospective studies were included. What was the timeline span for all these studies (started in what year? – 2008 to 2023). Discuss in Discussion section the changes in US technique over these 15 years. Was there improved Sp/Sens with more recent studies? Were TUS obtained at any time during the illness course, or at a specific number of days after start of illness? Were these one time TUS, or were there any repeat measurements on the same patient? Need more information.
5. Line 115 – should mention that there were 1570 patients without pneumonia, and 2866 with pneumonia dx. Was pneumonia diagnosis done based on what criteria? State that as well. Was TUS used as a criteria for pneumonia diagnosis, or were Chest x-ray, physical examination, symptoms or chest CT used to diagnose pneumonia in these 26 studies?
6. Line 115 – state how many males vs. females were included in the study (1897/1704), as there were more males - 52.6% vs. 47.6%. This needs to be discussed in the Discussion section, as could bring in bias.
7. Line 120 – instead of saying a majority of studies involved wards/ED, state the number 23, and 3 in the ICU (2 in NICU, 1 in PICU).
8. Need a patient characteristics summary/demographics table. This is missing currently.
9. Should not combine NICU and PICU, but keep those separated. Very different population and technique for ultrasounds.
10. Was there a chest x-ray done on all patients, the gold standard, to compare TUS findings to?
11. Must describe what each of the TUS findings are, and what they mean. What do consolidations look line on TUS – how can one tell one from the other apart? Put it in a table format in the Methods section. Explain what these are, and their significance (presence/absence is certain diseases):
a. Consolidation
b. Air bronchogram
c. Pleural effusion
d. B-line
e. Confluent B-lines?
f. Fluid bronchogram
g. Subpleural consolidation
h. Disappearance of lung sliding (isn’t this for pneumothorax?)
12. Based on Table 2, there were different types of probes used in all these studies (linear, convex, curvilinear). Should have a subgroup analysis on the impact of probe selection on specificity and sensitivity of TUS technique to diagnose pneumonia. Also, in discussion, must talk about the impact of probe selection – why one probe vs another, and why higher vs. lower MHx probe selection?
13. What does “limited ultrasound experience” mean in Table 2? Put this definition into methods.
14. Table 4 - what is U and what is L? Must put in footnotes of this table. Unclear what this table says with all the Ls and U, and the column headings. Why is this included?
15. Table 2 – only has 17 entry studies, but there were 26 in the study. Missing data, please correct!
16. Table 2 – what does ‘blinding’ refer to? Who was blinded and to what? Put this information in the methods section. While this table states that blinding was a YES on all studies, the text on page 10, line 10 says there were studies where interpreters were not blinded.
17. Subgroup analysis results should be placed into a table, with the end results of Specificity and Sensitivity based on ED vs. Ward vs NICU vs. PICU – calculate all these areas separately. Also need another table with Training level (limited, vs. experienced), with overall sensitivity and specificity.
18. Page 14_ Line 113 - must discuss what the sensitivity and specificity of chest x-ray is for pneumonia (86.2%, 98.2%, Balk et al, 2018), as that is gold standard currently. Compare new modality to gold standard. Again, including CXR data for included patients would be key, and compare TUS vs. CXR findings of pneumonia – a head to head comparison.
19. Page 15, line 170-179 – There is one outlier study, Zhan et al, which should be discussed here, as the residents there performed badly. Why was this? Should be discussed why there was such failure to use TUS in that study, and all the potential confounders.
20. Page 15, line 201 - Studies have already been done to assess the capacity of TUS to differentiate between viral and bacterial pneumonia (Elabbas et al, 2022).
21. Figure 5 adds very little to the point of this meta analysis, and likely can be omitted.
Minor Comments:
1. Title – make sure all words are capitalized (diagnosis, pneumonia, analysis)
2. Line 35- 36 – Accurate diagnosis of pneumonia in children is not challenging, and is very standard pediatric practice. This statement needs to be modified or deleted.
3. Table 1 - Column title “Gender” needs to be changed to “Sex”, as gender means something else.
4. Table 1 – has many NA for ages, and sex information. Why is there NA? Also, must write as footnote what abbreviations mean. Should state NA = ?.
5. Table 1 – totals of sample, males and females should be placed at the end of this table.
6. Table 3 – 5th column title should say “True” and not Ture.
Author Response
Please see the attachment.

------------------------------------------------------------
Reviewer 2 Report
Comments and Suggestions for Authors
Nice job. You can improve your work by:
1. You can talk about the differential diagnoses of pneumonia detected by US and add points on how to differentiate them. Or you can say we can only differentiate the pathologies based on clinical findings.
2. I recommend to use this article from Adil Elabbas : "Lung US beyond the diagnosis of pediatric pneumonia" with PMID of 35371734 which is relevant to your paper.
Author Response

(The authors gave the same response as above.)

Reviewer 3 Report
Comments and Suggestions for Authors
Several meta-analysis have already discussed the diagnostic value of thoracic ultrasound in diagnosing pediatric pneumonia. In particular, you can’t ignore the paper of Yan JH et al ( Lung ultrasound vs chest radiography in the diagnosis of children pneumonia: Systematic evidence. Medicine (Baltimore). 2020 Dec 11;99(50):e23671. doi: 10.1097/MD.0000000000023671. PMID: 33327356; PMCID: PMC7738074) which compares the diagnostic value of both TUS and chestXray.
Your discussion is mostly focused on the advantages of TUS (feasibility, no radiation exposure… ): a possible field of improvement could be TUS performed by pocket (hand-held) ultrasound devices.
Author Response
Please see the attachment.

====================================
Round 2
Reviewer 1 Report
Comments and Suggestions for Authors
The authors did a very nice job addressing all reviewer comments, and the manuscript is now in publishable form. The manuscript reads very well. No further comments.